# Fabrication and Potential Applications of Highly Durable Superhydrophobic Polyethylene Terephthalate Fabrics Produced by In-Situ Zinc Oxide (ZnO) Nanowires Deposition and Polydimethylsiloxane (PDMS) Packaging

**DOI:** 10.3390/polym12102333

**Published:** 2020-10-13

**Authors:** Jiangtao Hu, Mingxing Zhang, Yulong He, Maojiang Zhang, Rongfang Shen, Yumei Zhang, Minglei Wang, Guozhong Wu

**Affiliations:** 1State Key Laboratory for Modification of Chemical Fibers and Polymer Materials, Donghua University, Shanghai 201620, China; hujiangtao@sinap.ac.cn (J.H.); zhangym@dhu.edu.cn (Y.Z.); 2CAS Center for Excellence on TMSR Energy System, Shanghai Institute of Applied Physics, Chinese Academy of Sciences, No. 2019 Jialuo Road, Jiading District, Shanghai 201800, China; zhangmingxing@sinap.ac.cn (M.Z.); heyulong@sinap.ac.cn (Y.H.); zhangmaojiang@sinap.ac.cn (M.Z.); shenrongfang@sinap.ac.cn (R.S.); 3University of Chinese Academy of Sciences, Beijing 100049, China

**Keywords:** PET fabric, radiation-induced graft polymerization, ZnO nanowires, packaging

## Abstract

Considerable attention has been devoted to the in-situ deposition of zinc oxide (ZnO) nanowires (ZnO-NWs) on the surface of organic supports, due to their very wide applications in superhydrophobicity, UV shielding, and nanogenerators. However, the poor interfacial bond strength between ZnO-NWs and its support limits their applications. Herein, we developed a facile process to grow robust ZnO-NWs on a polyethylene terephthalate (PET) fabric surface through simultaneous radiation-induced graft polymerization, hydrothermal processing, and in-situ nano-packaging; the obtained materials were denoted as PDMS@ZnO-NWs@PET. The introduction of an adhesion and stress relief layer greatly improved the attachment of the ZnO-NWs to the support, especially when the material was subjected to extreme environment conditions of external friction forces, strong acidic or alkaline solutions, UV-irradiation and even washing with detergent for a long time. The PDMS@ZnO-NWs@PET material exhibited excellent UV resistance, superhydrophobicity, and durability. The ZnO-NWs retained on the fabric surface even after 30 cycles of accelerated washing. Therefore, this process can be widely applied as a universal approach to overcome the challenges associated with growing inorganic nanowires on polymeric support surfaces.

## 1. Introduction

The wetting behavior of a surface, as a macroscopic representation of liquid–solid interface interactions, has attracted considerable interest in both theoretical research and practical applications [1,2,3]. Superhydrophobic surfaces have recently become a popular topic because of their promising potential in a wide range of practical applications, such as sensing, solar cells, microfluidic devices, self-cleaning windows, and oil–water separation [4,5]. Generally, the surface wettability is governed by the combination of micro/nano-scale patterned structures and chemical compositions with low surface energy. Increasing the roughness and reducing the energy of the surface can significantly enhance its hydrophobicity [6]. In nature, the lotus leaf is a typical example of this superhydrophobic surface, whose self-cleaning process is referred to as the “lotus effect” [7].

Inspired by the lotus leaf, over the past few decades, superhydrophobic surfaces have usually been fabricated by first forming hierarchical micro/nanostructures and then coating the rough surface with a low-surface energy chemical compound [8,9]. Various nanomaterials were deposited on the surface of substrates to mimic the papilla of the lotus leaf, including zinc oxide (ZnO) [10], TiO_2_ [11], WO_3_ [12], silica [13], silver [14], and carbon [15]. ZnO is an interesting material because of its varied surface morphologies, including flower-like [16,17], nanorods [18,19,20,21], nanoneedles [22], and nanowires [23,24]. ZnO can be used to introduce different roughness sizes on the surface. Among numerous approaches for constructing micro/nano-scale structures, ZnO nanowires (ZnO-NWs) grown directly on substrates to ensure vertical alignment have received a lot of attention, due to their well-organized structure, high aspect ratio, and close arrangement, which generate a minimum liquid–solid contact fraction [25,26]. Moreover, ZnO-NWs with controlled morphology can be fabricated via a hydrothermal method and exhibit adjustable wettability after being modified with chemical compounds with different surface energies. However, the key factors that control the interfacial bond strength between the ZnO-NWs and their supports, affecting their durability in external harsh environments, are still not well understood. Moreover, the self-degradation of the organic substrate caused by the photo-generated electrons and holes of ZnO should also be carefully considered.

In order to improve the attachment between the ZnO crystal layer and a cotton fabric, some researchers added silane coupling agents in the pretreatment process to improve the tolerance and durability of the ZnO crystal layer in external harsh environments [27]. However, some shortcomings of this method must be emphasized. Firstly, the silicon hydroxyls produced by the hydrolysis of the silane coupling agent preferentially react with Zn-OH groups, thus reducing the reaction probability with the surface hydroxyls of the cellulose fiber. Secondly, many hydroxyl groups of the cotton fabric are wrapped in the crystal, resulting in a small amount of residual hydroxyl groups on the cellulose surface. Thirdly, the Si-O-C bond between the silane coupling agent and cellulose is sensitive to water, which leads to re-hydrolysis and failure of the bridging bonds in home-washing processes. Overall, the ZnO-NWs film peels off from the support surface when the as-prepared material is subjected to the repeated action of large deformations and external forces or strong acid/base conditions. Finally, the method lacks general applicability because the presence of hydroxyl groups on the trunk surface is a prerequisite for the formation of Si-O-C bonds between ZnO and its support.

Numerous methods were used to ameliorate the surface energy of polyethylene terephthalate (PET) fabrics, such as oxygen plasma treatment, chemical passivation etc. Ajmal et al. [28] treated the PET surface by using oxygen plasma to convert the surface state, which can improve the interface bonding strength. Zhou et al. [29] used chemical passivation to increase the surface roughness for promoting the attachment of ZnO nanolayers. However, the above methods cannot fundamentally solve the problem of ZnO peeling off from the surface of the organic carrier, especially in the case of large deformation or external force. Furthermore, no universal approach is available to improve the attachment of ZnO-NW arrays to their supports independently of the chemical nature of the polymers. Herein, we demonstrate the in-situ formation of ZnO-NWs on the surface of PET fabric as an example of this universal process. To enhance the adhesion strength between ZnO-NWs and organic support, a two-step procedure was adopted. First, γ-methacryloxypropyl trimethoxysilane (MAPS) was introduced on the surface of the PET fabric by radiation-induced graft polymerization (RIGP). After the surface modification, ZnO-NWs were deposited on the grafted fabric surface via a simple hydrothermal process, and a polydimethylsiloxane (PDMS) film was then formed on the ZnO-NWs surface, which improved not only their hydrophobicity but also their tolerance to external forces. The as-prepared material with hierarchical structures possessed superhydrophobic properties, facilitating the separation of oil–water mixtures. In order to assess the practical applicability of the materials, the durability of their surfaces was evaluated in various harsh environments, including accelerated washing (mechanical friction, detergent tolerance), UV irradiation, and strong acidic or alkaline solutions. In addition, the present fabrication approach can be extended as a universal approach to deposit ZnO-NWs onto the surface of different fabrics via the hydrothermal process and RIGP techniques.

## 2. Experimental

### 2.1. Materials

PET fabrics (plain weave, weight: 425 ± 20 g/m^2^) were purchased from Zhejiang Unifull Industrial Fibre Co., Ltd (Zhejiang, China). Before use, the fabric was extracted with acetone for 24 h to eliminate organic oil agents and dust on the surface. Sodium hydroxide (AR), MAPS (CR), acetone (AR), ethanol (AR), ZnC_4_H_6_O_4_•2H_2_O (AR), and hydroxyl-terminated polydimethylsiloxane (PDMS, CR) were provided by Sinopharm Chemical Reagent Co., Ltd. (China), and used as received.

### 2.2. Preparation of PET-g-PMAPS

First, the PET fabric was cleaned in acetone to eliminate surface impurities, and then cut into a size of 10 cm × 20 cm. The PET-*g*-PMAPS samples were fabricated by RIGP of MAPS. The reaction process is outlined in Scheme 1. In a previously published paper [30], the RIGP conditions, including absorbed dose, monomer concentration, reaction temperature, and reaction time, are described in detail.

### 2.3. Preparation of PDMS@ZnO-NWs@PET

The in-situ deposition of ZnO nanowires on the surface of the PET fabric was performed according to an improved process reported in the literature [31]. ZnO-NWs were formed in-situ on the PET fabric surface through the immobilization of ZnO seeds onto the fabric, followed by a hydrothermal process.

Modification of PET fabric with ZnO seeded solution. The first procedure was to prepare a ZnO seed solution using sodium hydroxide (0.120 g) and zinc acetate dihydrate (0.114 g). Zinc acetate dihydrate was dissolved in ethanol solution (100 mL) at 60 °C. Subsequently, sodium hydroxide in ethanol (75 mL) was added dropwise, and the obtained mixture was stirred at 60 °C for 3 h. After that, the PET fabrics were immersed in the ZnO seed solution for ~ 6 min and subsequently annealed at 100 °C for 30 min.

In-situ growth of ZnO-NW arrays on PET fabric surface. 6 g of zinc nitrate hexahydrate and 2.800 g of hexamethylenetetramine were dissolved in 200 mL deionized water, and the mixture was transferred into a Teflon-lined autoclave. First, 1 g of PET fabrics coated with ZnO seeds was placed in the Teflon-sealed autoclave, sealed, and heated at 95 °C for 4 h. Afterwards, the as-prepared sample was cleaned several times with ethanol and deionized water, and finally dried in a vacuum oven at 50 °C. The resulting product was denoted as ZnO-NWs@PET.

Hydrophobic treatment of PET fabric surface. Using ultrasounds, hydroxyl-terminated PDMS was dissolved in ethanol to prepare a 10 mM PDMS solution. To reduce the surface free energy of ZnO-NWs@PET, the samples were soaked in the PDMS ethanol solution for 30 min; then, the sample was dried to eliminate ethanol, followed by thermal treatment at 100 °C for 2 h to cure the PDMS layer. The as-obtained sample was denoted as PDMS@ZnO-NWs@PET.

### 2.4. Characterization

The chemical bonding characteristics of the different samples were compared using an attenuated total reflection infrared (ATR-IR) spectrometer (Bruker Tensor 207). Each curve was measured from 400 to 4000 cm^−1^ using an ATR apparatus with 4 cm^−1^ resolution.

The chemical elements on the surface of the different samples were identified by X-ray photoelectron spectroscopy (XPS, PHI-5702) from 0 to 1200 eV, using the C 1s peak at 285.0 eV as reference.

The crystalline phases of the substances deposited on the surface of the samples were characterized by X-ray diffraction (XRD, Rigaku D/MAX2200, Japan) with Cu/K_α_ radiation (λ = 1.54 Å).

The morphologies of the pristine PET fabric and functionalized PET fabrics were inspected by a field emission scanning electron microscopy (SEM, Merlin Compact, Zeiss, Jena, Germany) with Au-sputtered specimens.

The wetting properties of the pristine and functionalized PET fabrics were evaluated using a contact angle analyzer (KSV Instruments, Ltd., Finland) with 5 μL of test water. Six measurements were carried out at different locations on the surface of each sample, and the average value was used.

A certain amount of each sample was placed in a quartz crucible, and the relationship between the weight and temperature was determined by thermogravimetric (TG) analysis (TG 209 F3 Tarsus, Netzsch, Germany), using a heating rate of 10 °C/min in nitrogen atmosphere.

UV-vis absorption spectra of the original and functionalized PET fabrics were recorded using a Ruili 1100 spectrometer (Cary, NC, USA) in the range of 200 to 800 nm.

The oil content in the filtrate was evaluated using an infrared oil content analyzer (Oil 480, Beijing Chinainvent Instrument Tech., Co. Ltd., China), according to the following expression:(1)$$R\left(\%\right)\;=\;\left(1-\frac{C_{f}}{C_{i}}\right)\times 100$$
where *R*% represents the oil–water separation efficiency, while *C_i_* and *C_f_* are the oil concentrations in the oil–water mixture before and after separation, respectively.

Evaluation of washing durability: a laundering durability test was done on a standard color-fastness to washing laundering machine (Model SW12AII, Wenzhou Darong Textile Instrument Co., Ltd., Wenzhou, China). According to the American Association of Textile Chemists and Colorists (AATCC) Test Method 61-2006 no.2A, the PET fabrics were tailored to a size of 5 cm × 15 cm, and then the samples were washed in a rotating sealed stainless steel tank containing 50 stainless steel balls and 150 mL aqueous solution of WOB detergent (0.15%, w/w) in a thermostatically controlled water bath at 49 °C, 40 ± 2 rpm. One cycle of washing based on this standard program is equivalent to five cycles of home machine launderings, and the equal number of home machine washings is used in the present study.

UV radiation treatment: a 100-watt UV lamp was used to irradiate the pristine and modified fabric with a size of 5 cm × 5 cm in the wavelength range of 260–340 nm. The distance between the fabrics and the UV lamp was 10 cm.

## 3. Results and Discussion

### 3.1. Design and Characterization of PDMS@ZnO-NWs@PET

As discussed above, for most hydrophobic materials composed of an organic matrix and inorganic components, the micro/nanostructures on the surface are susceptible to mechanical damage under abrasion forces, due to their brittle nature and the poor adhesion strength between the inorganic component and its support. Therefore, a functional polymer film such as an organopolysiloxane can be introduced in the system as an adhesion and stress relief layer. The detailed preparation process is illustrated in Scheme 1. At variance with physically adhered polymer films prepared by solution-casting approaches, surface-grafted polymer chains exhibit a much stronger adhesion between the grafted chain end and the support surface, due to the formation of covalent bonds between the two components. In the present study, after grafting MAPS onto the surface of the pristine PET fabric, the siloxy groups were prone to hydrolyz to form silanols and react with the ZnO precursor, leading to a firmly attached ZnO seed layer on the surface of the PET fiber. However, under some conditions, the ZnO-NWs would still easily peel off, due to their brittleness and very high aspect ratio. The effectiveness of the hydroxyl-terminated PDMS is based on the following considerations: on the one hand, it endows the material with durable hydrophobicity, due to the low surface energy of PDMS and the reaction between the terminal silyl hydroxyl groups and ZnO; on the other hand, PDMS will infiltrate into the graft layer, due to its similar chemical structure and polarity. During the thermal curing process, the terminal silyl hydroxyl groups of PDMS will react with PMAPS or undergo self-polymerization to form a Si-O-Si flexible framework; thus, the ZnO-NWs will be encapsulated in-situ on the surface of the fiber. The formation of flexible films will slow down the damage to the micro/nanostructures on the surface induced by the harsh external environment, as shown in Scheme 1. This work also provides an approach for limiting the self-degradation of ZnO-coated organic supports due to the inherent photoactivity of ZnO, which has been described in detail in previous articles [30,32].

The variations in the chemical structure of the fabrics induced by the surface modification can be examined through the ATR-IR spectroscopy analysis, as given in Figure 1a. Compared with the spectrum of the pristine PET fabric, the PET-*g*-PMAPS curve exhibits a new peak assigned to Si-C (815 cm^−1^) [33], confirming that MAPS was successfully grafted on the PET fabric surface. The spectrum of ZnO-NWs@PET shows two new peaks at 430 cm^−1^ and 465 cm^−1^, corresponding to the absorption of Zn-O-Zn and Zn-O-Si bonds, respectively [34], indicating that the ZnO-NWs were anchored in-situ on the surface of the PET fabric using the PMAPS grafted chains as a bridge.

XPS is an effective characterization method for determining the chemical composition of a material [35]. Wide-scan XPS spectra were obtained to analyze the chemical compositions of the pristine and modified PET fabrics. The corresponding results are shown in Figure 1b. After grafting, Si 2s and Si 2p peaks were observed at approximately 152.7 and 100.9 eV, respectively, revealing that the PMAPS molecules were anchored onto the surface of the PET fiber via chemical bonds [30]. Upon coating with the ZnO-NWs, the characteristic Zn 2p, Zn 3s, and Zn 3p peaks were detected in the spectrum, denoting the presence of ZnO-NWs on the fabric; however, the Si 2s and Si 2p peaks disappeared, due to the coverage of the ZnO seed layer and ZnO-NWs. In the spectrum of PDMS@ZnO-NWs@PET, the intensity of the C 1s peak increased, whereas that of the Zn peaks was greatly reduced. These changes can be attributed to the etherification between the Si-OH groups of PDMS and the Zn-OH species, which leads to the hydrophobic PDMS film firmly adhering to the surface of the ZnO-NWs. To further confirm the formation of chemical bonds among ZnO-NWs, PDMS layer, and MAPS grafted chains, the O 1s core-level spectra were provided. Based on the O 1s core-level spectra (Figure 1c), the oxygen atoms on the PDMS@ZnO-NWs@PET surface were present in the form of Zn-O-Zn, Si-O-Zn, -OH, and -Si-O-Si- bonds, indicating the formation of chemical bonds among different component layers [36]. These results are consistent with the FTIR analysis.

Figure 1d shows the XRD patterns of PET, PET-*g*-PMAPS, ZnO-NWs@PET, and PDMS@ZnO-NWs@PET. PET and PET-*g*-PMAPS exhibited similar patterns, indicating that the RIGP process did not damage the crystal structure of the PET support. The spectra of ZnO-NWs@PET and PDMS@ZnO-NWs@PET show two sets of diffraction peaks, one for the PET support and the other for the ZnO-NWs, demonstrating that the as-synthesized samples are well-constructed. A series of peaks belonging to crystalline ZnO with wurtzite microstructure were observed at 2*θ* values of 31.5° (100), 34.5° (002), 36.5° (101), 47.8° (102), 56.7° (110), 62.8° (103), and 68.1° (211), which are consistent with the data list for JCPDS, card no. 36-1451 [37]. In addition, the ZnO-NWs clearly exhibited a preferred (101) orientation. It is well-known that when crystal grains grow along the (101) plane, they usually exhibit elongated triangular-shaped strips, which is consistent with the SEM results (Figure 2e) [38].

Figure 2 shows representative SEM images of the PET fabric (a), PET-*g*-PMAPS (b), ZnO-NWs@PET (c, e), and PDMS@ZnO-NWs@PET (d, f) samples. It can be clearly seen that the pristine PET fabric had a smooth and clean surface (Figure 2a), and a similar morphology was also observed for PET-*g*-PMAPS, indicating that the RIGP treatment caused slight or no damage to the PET fabric. After the hydrothermal reaction, dense ZnO-NWs grew radially around a single PET fiber axis and formed a “nanobrush” structure (Figure 2c). The obtained sample was highly flexible; however, after a large deformation, the ZnO-NWs were peeled off from the surface of the grafted PET fabric (Figure 2e). After coating with PDMS (Figure 2d), the morphology of the ZnO-NWs changed significantly. In particular, the lower part of the nanowires was covered by PDMS, while ZnO retained its nanowire morphology, ensuring a high specific surface area. In addition, the average diameter of the ZnO-NWs also showed a slight increase, indicating the successful in-situ packaging of ZnO-NWs with PDMS at the nanoscale.

The durability (especially service life and recyclability) of functional fabrics is an important factor that affects their practical applicability. A wash resistance test was thus performed to evaluate the reusability of the present samples. Washing is a complex process that involves chemical interactions between the detergent and the fabric, as well as mechanical interactions such as friction and shear with steel balls, container wall, and water (Figure 3a,b). After 30 accelerated washing cycles, the ZnO-NWs still uniformly wrapped the surface of the PET fabric (Figure 2f), and no abscission phenomenon was observed. The durability was further evaluated by performing water contact angle (WCA) measurements after 30 laundering cycles (inset of Figure 2d,f). The contact angle values did not change significantly (155.5° vs. 154.2°). The superhydrophobic characteristics of PDMS@ZnO-NWs@PET can be attributed to the high roughness of the ZnO-NWs and the low surface free energy of the PDMS coating.

The durability and robustness of the coatings were further assessed by monitoring variations in the hydrophobicity of the obtained samples upon immersion in aqueous solutions of different pH. The PDMS@ZnO-NWs@PET sample showed higher resistance to strongly acidic media, but slightly weaker resistance to strongly alkaline environments. In particular, after the immersion of the obtained sample in an acidic solution, the WCAs increased slightly as a result of further condensation between Si-OH and Zn-OH, which effectively reduced the amount of polar group. However, upon extending the soaking period, the ester bond in the modified fabric was broken, and the density of polar groups increased. Ester bonds are more easily broken under alkaline than acidic conditions [39]. For example, under alkaline condition, the PET matrix was rapidly hydrolyzed to produce super-hydrophilic ethylene glycol and disodium terephthalate salt (Scheme 2), which greatly impairs the water-repellency performance of the resultant.

In order to test the change in hydrophobic properties of the material when used outdoors, the modified fabrics were irradiated with a 100 W UV-lamp. Under UV light irradiation, the contact angle of the modified fabric did not change significantly, indicating that the superhydrophobicity of the modified fabric was very stable. This phenomenon may be ascribed to the excellent UV absorption capacity of the ZnO-NWs and the stable chemical structure of the interface layer between the hydrophobic coating and the matrix, which has been discussed in our previous studies [32,40]. Overall, the water-repellent properties of the obtained samples enabled them to maintain a high contact angle under different extreme conditions. The excellent resistance to external factors is primarily ascribed to the formation of a three-dimensional net pocket structure involving PDMA, PMAS, and ZnO-NWs.

As most hydrophobic materials are often used outdoors, their UV radiation resistance is another important property. A UV-Vis test was performed to study the UV-resistance of the pristine and modified fabrics. As shown in Figure 4, the pristine PET fabric exhibits a high absorption peak in the ultraviolet region, indicating that the PET fabric was easily degraded because of its high UV absorption capacity. In addition, the introduction of the grafted chains did not improve the UV resistance of the PET fabric. After the in-situ formation of ZnO-NWs on their surface, ZnO-NWs@PET fabrics exhibited higher UV absorption capacity because of the presence of the ZnO-NWs [16], which can prevent UV-induced damage to the fibers. In addition to improving the hydrophobic properties of the obtained-material, another reason for using PDMS is that the bond energy of Si-O-Si is as high as 460 kJ/moL, which is much higher than the UV irradiation energy (333–406 kJ/mol) [41]. Therefore, the material will not lose its hydrophobic properties during its usage outdoors. Furthermore, an important characteristic associated with the use of MAPS as a bridge structure must be emphasized. It can prevent the photocatalytic self-degradation of the PET trunk by the ZnO-NWs, due to the presence of an organic–inorganic crosslinking network formed by the hydrolysis of the grafted PMAPS chains.

Figure 5 shows the TG and derivative TG (DTG) curves of the pristine and modified PET fabrics. The corresponding thermal decomposition parameters, including the initial decomposition temperature (*T_di_*), temperature of maximum degradation rate (*T_max_*), and char yield (*Y_c_*) at 800 °C, are listed in Table 1. Due to the low molecular weight and poor heat resistance of the grafted chains, when MAPS was grafted onto the surface of the PET fabric, the *T_di_* value of PET-*g*-PMAPS reduced by approximately 129 °C. After the in-situ deposition of ZnO-NWs, the *T_di_* value increased by nearly 100 °C. This phenomenon may be ascribed to the following causes. During the in-situ formation of ZnO-NWs on the PET surface, the grafted chains will react with the ZnO precursor to form a 3D crosslinked network containing Si-O-Zn and Si-O-Si bonds, thus improving the thermal resistance of the grafted chains. During the degradation process, MAPS/ZnO-NWs generates a dense inorganic layer composed of SiO_2_ and ZnO, which, to some extent, can isolate the fibers from air and inhibit the degradation of the inner part. After the hydrophobic treatment of the ZnO-NWs, the *T_di_* value decreased again, due to the low thermal resistance of PDMS. Based on the analysis of the DTG curves, the thermal degradation process changed from the one-step process of the original PET fabric to a multi-step degradation of the modified fabrics. Furthermore, DTG peaks with reduced heights and varieties of *Y*_c_ were also observed for the modified fabrics, demonstrating that the functionalized fabrics possessed a different degradation mechanism. The specific reasons for this phenomenon were discussed in the structure analysis.

### 3.2. Oil–water Separation Behaviors of PDMS@ZnO-NWs@PET

Following the analysis of the hydrophobic properties discussed above, the oil–water separation ability was studied by incorporating the PDMS@ZnO-NWs@PET fabric in a model device (Figure 6a). A mixture of chloroform (dyed with Sudan Ⅲ) and water (dyed with methylene blue for easy observation) in 1:1 volumetric ratio was poured onto the PDMS@ZnO-NWs@PET fabric. Due to the superhydrophobicity-superoleophilicity characteristics and as chloroform is denser than water, only chloroform could pass through the fabric and reach the underlying container, while water was blocked by the separation material. The separation process was solely driven by gravity, and the same process can be used for other weight oils with a higher density than water. To further assess the durability of the PDMS@ZnO-NWs@PET fabric, we examined its cycling separation efficiency. As shown in Figure 5b, the fabric retained a high separation capacity even after 10 cycles, demonstrating the superior recyclability of the fabric.

As a typical superhydrophobic surface, the self-cleaning properties of PDMS@ZnO-NWs@PET were confirmed by assessing its removal ability of model solid pollutants. As shown in Figure 6c,d, simulated dust (CuSO_4_•5H_2_O) deposited on the surface of PDMS@ZnO-NWs@PET could be easily eliminated after washing with water. The self-cleaning ability of PDMS@ZnO-NWs@PET derives from the protection of the lubricant layer, which isolates the substrate from the test solid. Furthermore, the high roughness and hydrophobicity of the coating results in an ultralow contact fraction between PDMS@ZnO-NWs@PET and water, which enables the latter to easily slide off even at a very small tilt angle.

### 3.3. Other Potential Applications of In-Situ Nano-Packaging Technology

As a preliminary investigation, this study focuses only on the traditional hydrophobic applications of fiber materials loaded with inorganic nanowires. However, the preparation process can be readily extended to fabricate wearable electronics, fiber-based generators, strain sensors, etc., As shown in Scheme 3, we prepared a strain sensor based on elastic PET fabrics coated with Ag@ZnO-NWs and a durable multifunctional detector based on Fe_3_O_4_-NWs and polypropylene fabrics to monitor variations in UV and temperature conditions for proof-of-concept purposes. The present process not only improves the durability of semiconductors and their doping components but can also achieve effective doping on heterogeneous surfaces. Relevant work is in progress and will be published in the future.

## 4. Conclusions

In summary, we introduced a reliable approach for the fabrication of PDMS@ZnO-NWs@PET hybrid fabrics via radiation-induced graft polymerization of MAPS and subsequent deposition of ZnO nanowires, followed by in-situ packaging. The grafted chains played four roles: (i) improving the adhesion of the ZnO-NWs to the support, (ii) creating a stress buffer layer with three-dimensional structure via the reaction between MAPS, PDMS, and ZnO-NWs, (iii) forming an organic–inorganic hybrid layer to prevent photocatalytic self-degradation of the support, and (iv) enhancing the superhydrophobilicity of the resulting material. PDMS@ZnO-NWs@PET showed excellent UV resistance and chemical stability upon immersion in strongly acidic and alkaline solutions for a long time. In addition, the modified fabric showed effective separation of oil–water mixtures, excellent recycling performance, and self-cleaning properties, making it a promising candidate for potential application in sewage purification. In principle, this approach can be extended as a universal method for the in-situ deposition of various inorganic nanowires on an organic support surface with ultra-high durability and robustness.

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
