# Peer review of "Fabrication and Potential Applications of Highly Durable Superhydrophobic Polyethylene Terephthalate Fabrics Produced by In-Situ Zinc Oxide (ZnO) Nanowires Deposition and Polydimethylsiloxane (PDMS) Packaging"

_polymers, 2020, doi:10.3390/polym12102333_

Round 1
Reviewer 1 Report
The manuscript " Fabrication and Potential Applications of Highly Durable Superhydrophobic Polyethylene Terephthalate Fabrics Produced By In Situ Zno Nanowires Deposition And Polydimethylsiloxane Packaging" by Minglei Wang et al. deals with the elaboration and characterization of ZnO nanowires on PET and covered with PDMS. The topic seems interesting and actual, but I do have some suggestions as to the content of the manuscript before finalizing the publication of this manuscript in Polymer.
To begin with, there are some mistakes in the English language, typos and missing texts are observed in their write-up, for example, In the title (1) Zno. (2) they used PDMS in the abstract part without defining the term. Although, PDMS is a very common representation. However, they have to write their complete form as they mentioned for PET. It also needed to correct in the introduction part too. (3) please use one representation for micro-nano either micro/nano or micro-nano etc. in the whole text. (4) one-time, they have used term ZnO and in the following text, they are writing it as zinc oxide (page #2, after reference # 15). Similarly, there are un-necessary full stops (page # 4, scheme.1. etc.) (5) additional spaces between the connected words as hydroxyl-. (6) please use one notation for figure captions either as Figure or Fig. See the caption of Figure 1 and look carefully at where to put the full stops. (7) there is no Figure # for the figure on page #7. (8) define PP fabrics. Please read as many times as you can of the revised version before your next-submission.
- Q # 1: It is suggested to include a few other methods of PET surface treatment such as oxygen plasma treatment along with chemical passivation method as you already mentioned to adjust the surface energy for a more comprehensive review on the PET for the growth of ZnO-nano-structures, refers to some recently published reports like; 3390/nano10061225, 10.3390/nano9081067.
- Q #2: I would recommend you write the thickness of the PDMS passivation layer on ZnO-NW under subsection 2.3 of the experimental part or somewhere else.
- Q #3: Please check the text in the schematic illustration. Choose the appropriate color, position, and size of the text in the figures. Same for Figure #2 and check the consistency of text size and style in the axis.
- Q #4: please write the wide-range of XPS in the initial paragraph of the XPS explanation on page #5. And explain more clearly why carbon peaks have increased significantly in PDMS@ZnO-NWs@PET and PET-g-PMAPS if you have not mentioned it, or I missed it. And in Figure 1c, I am not getting your point either you want to explain the stoichiometric contents of oxygen or something else.
- Q #5: please write the dimensions of PET fabrics if not mentioned already.
- Q #6: they mentioned about durability and robustness tests of the coatings upon immersion in aqueous solutions of different pH and samples showed relatively weaker resistance under alkaline environments; please write the time-span for immersion in the solution and explain the possible cause of it, is it because of ester bonds breakages? if yes, write it clearly.
- Q #7: About UV light irradiation test, they have not mentioned the UV intensity, how long they exposed samples under UV and the sample area at which UV was irradiated?
- Q #8: How about the PDMS damage capacity under UV? Please include this test as well if you have examined it because this test seems important for your study and it can increase the reliability of your conducted research.
- How about the UV-Vis absorption spectra of PDMS? Have you conducted this test?
- If I have not missed the information, would you mind pointing out about the experimental evidence of 30 cycles of washing test as mentioned in the abstract?
- Would you mind replacing the word flexible in the conclusion part either with transparent, polymer, or plastic if you are not including any flexibility test here?
Please check the reference style, especially Ref. 18, 38, etc.
Author Response
To begin with, there are some mistakes in the English language, typos and missing texts are observed in their write-up, for example, In the title (1) ZnO. (2) they used PDMS in the abstract part without defining the term. Although, PDMS is a very common representation. However, they have to write their complete form as they mentioned for PET. It also needed to correct in the introduction part too. (3) please use one representation for micro-nano either micro/nano or micro-nano etc. in the whole text. (4) one-time, they have used term ZnO and in the following text, they are writing it as zinc oxide (page #2, after reference # 15). Similarly, there are un-necessary full stops (page # 4, scheme.1. etc.) (5) additional spaces between the connected words as hydroxyl-. (6) please use one notation for figure captions either as Figure or Fig. See the caption of Figure 1 and look carefully at where to put the full stops. (7) there is no Figure # for the figure on page #7. (8) define PP fabrics. Please read as many times as you can of the revised version before your next-submission.
Ok, the paper has been carefully polished in the revised manuscript.
Q # 1: It is suggested to include a few other methods of PET surface treatment such as oxygen plasma treatment along with chemical passivation method as you already mentioned to adjust the surface energy for a more comprehensive review on the PET for the growth of ZnO-nano-structures, refers to some recently published reports like; 3390/nano10061225, 10.3390/nano9081067.
Ok, a few common methods of PET surface treatment have been added in the revised manuscript. In detail, “Numerous methods were used to ameliorate the surface energy of PET fabrics, such as the oxygen plasma treatment, chemical passivation etc. Ajmal et al treated PET surface by oxygen plasma to convert the surface state, which can improve interface bonding strength. Zhou et al used chemical passivation to increase surface roughness for promoting the attachment of ZnO nanolayers. However, the above methods can not fundamentally solve the problem of ZnO peeling off from the surface of organic carrier, especially in the case of large deformation or external force. Furthermore, no universal approach is available to improve the attachment of ZnO-NW arrays to their supports independently of the chemical nature of the polymers.”
Q #2: I would recommend you write the thickness of the PDMS passivation layer on ZnO-NW under subsection 2.3 of the experimental part or somewhere else.
Thank you for your suggestion. Because the ZnO-NW are firmly loaded on the surface of the fiber, we cannot separate a single ZnO-NW coated with PDMS to measure its diameter.
Q #3: Please check the text in the schematic illustration. Choose the appropriate color, position, and size of the text in the figures. Same for Figure #2 and check the of text size and style in the axis.
Ok, the color, position, and size of the test in Scheme. 1 have been re-adjusted to make the graphics more recognizable. In Fig. 2, the text size and style in the axis have been checked and modified to maintain consistency.
Q #4: please write the wide-range of XPS in the initial paragraph of the XPS explanation on page #5. And explain more clearly why carbon peaks have increased significantly in PDMS@ZnO-NWs@PET and PET-g-PMAPS if you have not mentioned it, or I missed it. And in Figure 1c, I am not getting your point either you want to explain the stoichiometric contents of oxygen or something else.
The wide-range has been added in characterization section.
Awfully sorry for this mistake, we have rearranged X-ray photoelectron spectroscopy by normalization.
To prove the formation of chemical bonds between the functional coating and their organic carrier, the O 1s core-level spectra were provided. In detail, “To further confirm the formation of chemical bonds among ZnO-NWs, PDMS layer, and MAPS grafted chains, the O 1s core-level spectra were provided. Based on the O 1s core-level spectra (Fig. 1c), the oxygen atoms on the PDMS@ZnO-NWs@PET surface were present in the form of Zn-O-Zn, Si-O-Zn, -OH, and -Si-O-Si- bonds, indicating the formation of chemical bonds among different component layers [36]; these results are consistent with the FTIR analysis.”.
Q #5: please write the dimensions of PET fabrics if not mentioned already.
Ok, the dimensions of PET fabrics have been added in experimental section. In detail, “First, the PET fabric was cleaned in acetone to eliminate surface impurities, and then cut it into a size of 10 cm × 20 cm.”.
Q #6: they mentioned about durability and robustness tests of the coatings upon immersion in aqueous solutions of different pH and samples showed relatively weaker resistance under alkaline environments; please write the time-span for immersion in the solution and explain the possible cause of it, is it because of ester bonds breakages? if yes, write it clearly.
Ok, the time-spans for immersion in the different solutions have been provided in Fig. 3c. We think that the decrease in contact angle is caused by hydrolysis under acid or alkali conditions. In detail, “Ester bonds are more easily broken under alkaline than acidic conditions[1]. For example, under alkaline condition, the PET matrix was rapidly hydrolyzed to produce super-hydrophilic ethylene glycol and disodium terephthalate salt (Scheme. 2), which greatly impairs the water-repellency performance of the resultant.’’
Scheme. 2 The mechanism of alkaline hydrolysis of PET.”.
Q #7: About UV light irradiation test, they have not mentioned the UV intensity, how long they exposed samples under UV and the sample area at which UV was irradiated?
Ok, some necessary information has been supplemented in experimental section and Fig. 3c. In detail, “UV Radiation Treatment: A 100-watt UV lamp was used to irradiate the pristine and modified fabric with a size of 5 cm ×5 cm in the wavelength range of 260-340 nm. The distance between the fabrics and the UV lamp was 10 cm.”.
Q #8: How about the PDMS damage capacity under UV? Please include this test as well if you have examined it because this test seems important for your study and it can increase the reliability of your conducted research.
There is no separate UV resistance test for PDMS, but we believe that PDMS has good UV resistance. The material that is not coated with PDMS exhibits hydrophilic properties, while after coating with PDMS, the material becomes superhydrophobic. During the entire ultraviolet irradiation treatment, the hydrophobic properties of the material did not change significantly, indicating that the PDMS coating did not undergo significant irradiation degradation. In detail, “In addition to improving the hydrophobic properties of the obtained-material, another reason for using PDMS is that the bond energy of Si-O-Si is as high as 460 kJ/mol, which is much higher than the UV irradiation energy (333–406 kJ/mol)[2]. Therefore, the material will not lose its hydrophobic properties during its usage in outdoor.”.
How about the UV-Vis absorption spectra of PDMS?[1] Have you conducted this test?
Ok, UV-Vis absorption spectrum of PDMS has been provided in Fig. 4.
Fig. 4. UV-vis absorption spectra of pristine PET fabric, PET-g-PMAPS, ZnO-NWs@PET, PDMS, and PDMS@ZnO-NWs@PET.
If I have not missed the information, would you mind pointing out about the experimental evidence of 30 cycles of washing test as mentioned in the abstract?
Ok, the experimental evidence of 30 cycles of washing test was provided in Fig. 3. In detail, “A wash resistance test was thus performed to evaluate the reusability of the present samples. Washing is a complex process that involves chemical interactions between the detergent and the fabric, as well as mechanical interactions such as friction and shear with steel balls, container wall, and water (Fig. 3a, b). After 30 accelerated washing cycles, the ZnO-NWs still uniformly wrapped the surface of the PET fabric (Fig. 2f), and no abscission phenomenon was observed. The durability was further evaluated by performing water contact angle (WCA) measurements after 30 laundering cycles (inset of Figs. 2d, f). The contact angle values did not change significantly (155.5o vs 154.2o). The superhydrophobic characteristics of PDMS@ZnO-NWs@PET can be attributed to the high roughness of the ZnO-NWs and the low surface free energy of the PDMS coating.” Moreover, the experimental method of accelerated washing is also supplemented in the experimental part. In detail, “Evaluation of washing durability: A laundering durability test was done on a standard color-fastness to washing laundering machine (Model SW12AII, Wenzhou Darong Textile Instrument Co., Ltd., China). According to the AATCC (American Association of Textile Chemists and Colorists) Test Method 61-2006 no.2A, the PET fabrics were tailored to a size of 5 cm × 15 cm, and then the samples were washed in a rotating sealed stainless steel tank containing 50 stainless steel balls and 150 mL aqueous solution of WOB detergent (0.15%, w/w) in a thermostatically controlled water bath at 49 °C, 40 ± 2 rpm. One cycle of washing based on this standard program is equivalent to five cycles of home machine launderings, and the equal number of home machine washings is used in present study.”.
Would you mind replacing the word flexible in the conclusion part either with transparent, polymer, or plastic if you are not including any flexibility test here?
Ok, the word flexible in the conclusion section has been deleted in the revised manuscript.
Please check the reference style, especially Ref. 18, 38, etc.
Ok, all reference formats have been carefully checked.
[1] Kumar, S. and C. Guria, Alkaline Hydrolysis of Waste Poly(Ethylene Terephthalate): A Modified Shrinking Core Model. J. Macromol Sci A 2005, 42 (3), 237-251.
[2] Zhu, X.L., et al., Unique surface modified aramid fibers with improved flame retardancy, tensile properties, surface activity and UV-resistance through in situ formation of hyperbranched polysiloxane-Ce0.8Ca0.2O1.8 hybrids. J. Mater. Chem. A 2015, 3 (23), 12515-12529.
Reviewer 2 Report
1) I recommend the author to add SEM micrograph for the cross-sectional view of the variants of NWs.
2) Author have mentioned about the diameter but why the author has not shown the magnified view of a single NW?
Author Response
1) I recommend the author to add SEM micrograph for the cross-sectional view of the variants of NWs.
Thank you for your suggestion. Due to the current summer vacation, the testing institutions and students are on their vacation, so we cannot provide the SEM micrographs for the cross-sectional view of the variants of NWs at present situation.
2) Author have mentioned about the diameter but why the author has not shown the magnified view of a single NW?
The size of nanowires is the size conversion of nanowires on the surface of the resultant according to the scale. In order to ensure the accuracy of the expression, we delete the description about the size of nanowires in this paper.
Round 2
Reviewer 1 Report
The authors have improved the manuscript a lot. It is almost ready for the publication, however, I would have a suggestion for the authors if they want to follow it. This is up to them. please look at the labels (format size, figure no. 1) of all figures if they are even. This is for their benefit, your paper presentation will be more attractive for the readers if you will consider it,
Author Response
Ok, some necessary changes have been made and are marked in blue in the text.
Reviewer 2 Report
I would like to thank the author for responding to my comments. However, the core concept of your manuscript can't be justified without the cross-sectional morphology of the fabricated Nano-wires. I am not doubting your results, but the manuscript will be more presentable. I suggest to add the SEM micrographs for the cross sectional view of the NWs.
Thank you.
Author Response
Thanks for your suggestion. We have made many attempts to prepare cross-sectional samples that can be easily observed. However, due to the different properties of the PET matrix, graft layer, ZnO-nanowire and PDMS coating, it is difficult to cut a flat cross-section of the composite for observation, which makes it difficult to focus in the process of taking electron microscopy. So far, we have not found a feasible method to prepare the cross-sectional samples of this composite for SEM observation. The cross-sectional photo of the composite we tried to take is shown below.